# MiLDEdit: Reasoning-Based Multi-Layer Design Document Editing

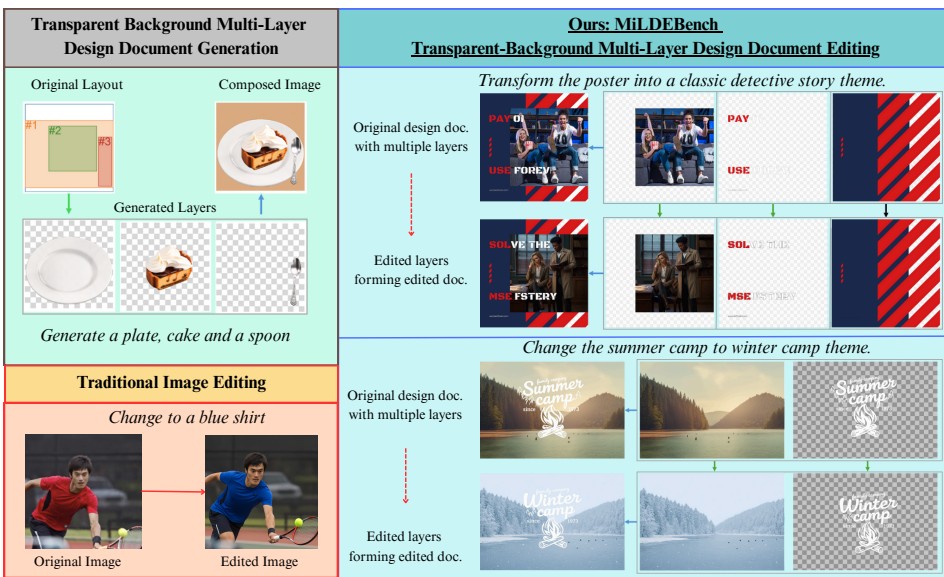

Figure 1: Examples of MiLDEBench. Different from traditional image editing and multi-layer generation task, our benchmark is the first targeting to transparent-background, multi-layer design document editing.

## ABSTRACT

Real-world *design documents* (e.g., posters) are inherently multi-layered, combining decoration, text, and images. Editing them from natural-language instructions requires fine-grained, layer-aware reasoning to identify relevant layers and coordinate modifications. Prior work largely overlooks *multi-layer design document editing*, focusing instead on single-layer image editing or multi-layer generation, which assume a flat canvas and lack the reasoning needed to determine *what* and *where* to modify. To address this gap, we introduce the Multi-Layer Document Editing Agent (**MiLDEAgent**), a reasoning-based framework that combines an RL-trained multimodal reasoner for layer-wise understanding with an image editor for targeted modifications. To systematically benchmark this setting, we introduce the Multi-Layer Document Editing Benchmark (**MiLDEBench**), a human-in-the-loop corpus of over 20K design documents paired with diverse editing instructions. The benchmark covers both *content* and *layout* edits and is complemented by a task-specific evaluation protocol, **MiLDEEval**, which spans four dimensions for content editing (instruction following, layout consistency, aesthetics, and text rendering) and two dimensions for layout editing (instruction following and content consistency). Extensive experiments on 14 open-source and 2 closed-source models reveal that existing approaches fail to generalize: open-source models often cannot complete multi-layer document editing tasks, while closed-source models suffer from instruction misalignment and format violations. In contrast, MiLDEAgent achieves strong layer-aware reasoning and precise editing, delivering over 50% improvements compared to all open-source baselines and attaining performance comparable to closed-source models, thereby establishing the first strong baseline for multi-layer document editing.

# 1 INTRODUCTION

While recent breakthroughs in image generation have transformed creative workflows, editing real-world design documents such as posters, flyers, and slides still remains an open challenge. Unlike natural images, these design documents are intrinsically multi-layered, combining backgrounds, graphics, text, and foreground imagery in a carefully structured hierarchy. Effective editing requires reasoning about which layers are relevant to user intent, how their relationships constrain possible modifications, and where changes can be applied without disrupting layout or occluding critical content. Existing reasoning-based editing methods (Jiang et al., 2025; Chen et al., 2025b; Zhang et al., 2025b) are built for flat, single-layer canvases and fail to capture this complexity. Besides, despite some works focuing on design document generation workflows (Huang et al., 2024; Pu et al., 2025; Chen et al., 2025a), layer-aware document editing remains largely unexplored, leaving a critical gap in vision-language reasoning and multimodal editing.

To fill this gap, we propose the first benchmark for reasoning-based *multi-layer* document editing, Multi-Layer Document Editing Benchmark (**MiLDEBench**). MiLDEBench systematically targets two complementary axes: *content editing*, which demands semantic modification while preserving layout coherence, and *layout editing*, which requires geometric manipulation under inter-layer constraints. Building on 20K transparent-background documents from the public Crello dataset (Yamaguchi, 2021), we synthesize 50K natural-language editing instructions and 87K layer-aligned edit steps through a hybrid pipeline that combines open-source multimodal LLMs with human-in-the-loop verification. For content editing, we aim to approximate real-world application scenarios where users come from diverse backgrounds. To this end, we design persona-conditioned and document-conditioned prompts that capture heterogeneous editing intents, ensuring that the dataset reflects a broad spectrum of user needs (*e.g.*, converting a Christmas card into a Halloween card). While for layout editing, we instantiate four fundamental layer-level primitives—*move*, *swap*, *scale*, and *rotate*.

To evaluate this new setting, we introduce **MiLDEEval**, a task-specific protocol covering four content dimensions: *instruction following*, *layout consistency*, *aesthetics*, and *text rendering*, as well as two layout dimensions: *instruction following* and *content consistency*. Together, these components provide a standardized testbed for reasoning-intensive, layer-aware editing that reflects practical multi-layer design document editing scenarios. We evaluate 14 open-source and 2 closed-source image-editing models on MiLDEEval. Open-source models show limited instruction-following capability, often returning unmodified documents, while closed-source models achieve higher semantic alignment and visual quality but sometimes compromise layout or format consistency. Incorporating explicit reasoning yields only modest improvements, indicating that existing reasoning modules are largely text-centric and do not fully leverage the multi-layer document structure. These findings suggest that multi-layer design document editing poses challenges beyond the scope of current image-editing paradigms and motivate the need for a reasoning-based, layer-aware approach.

To address these limitations, we propose **MiLDEAgent**, a reasoning-based, layer-aware editing agent. It integrates (i) an RL-trained multimodal reasoner, optimized with a novelly designed reward for layer identification and layer-conditioned editing prompt synthesis, and (ii) a pluggable image editor for targeted layer-wise modifications. Experimental results on content editing task indicate that explicit layer-aware reasoning is critical for faithful and controllable document-level editing. Our agent outperforms all open-source models by over 50%, achieves instruction-following performance comparable to closed-source models, and surpasses them in layout consistency. Notably, it achieves the best balance between *instruction adherence* and *layout consistency*, highlighting the effectiveness of reasoning-based multi-layer editing.

We summarize our main contribution as follows:

- **Task and Benchmark.** We formalize the problem of *multi-layer design document editing* and introduce **MiLDEBench**, a corpus of 20K documents with 50K editing instructions and 87K layer-aligned steps spanning both *content* and *layout* edits, along with the task-specific evaluation protocol **MiLDEEval**.

- **Comprehensive Evaluation.** We benchmark 14 open-source and 2 closed-source systems, identifying consistent challenges in instruction following, layout fidelity, and coordination across layers.

- **Method and Results.** We propose **MiLDEAgent**, which combines a GRPO-trained multimodal reasoner with a pluggable layer-wise editor. MiLDEAgent demonstrates strong instruction adherence and layout consistency, surpassing open-source baselines and performing competitively with closed-source systems.

## 2 RELATED WORK

**Multi-Layer Image Generation.** Existing work on multi-layer design documents has predominantly focused on *generation*. Datasets for this purpose are typically constructed by harvesting layered assets from large-scale corpora (*e.g.*, LAION (Schuhmann et al., 2022), COCO (Lin et al., 2014)) (Zhang et al., 2023b; Huang et al., 2024; Tudosiu et al., 2024; Gu et al., 2024) or by curating poster- and graphic-style designs from content platforms (Pu et al., 2025; Yamaguchi, 2021). Concurrent studies explore unified models that jointly perform generation and understanding with reasoning capabilities (Deng et al., 2025; Xiao et al., 2025b), as well as synthetic data pipelines (Chen et al., 2025a; Burgert et al., 2024), while several frameworks explicitly aim to produce coordinated multi-layer outputs (Huang et al., 2024; Pu et al., 2025; Chen et al., 2025a). In contrast, practical design workflows are often characterized by non-expert users editing *existing* documents under high-level instructions while preserving structure. This omission creates a clear gap between current research and real-world needs. To bridge this gap, we introduce **MiLDEBench**, the first benchmark that pairs layered documents with document-level instructions and stepwise, layer-aligned edit traces validated through human evaluation, reframing the task from multi-layer generation to faithful and controllable multi-layer editing.

**Reasoning-based Image Generation and Editing.** Driven by recent advances in large language models (LLMs) and training algorithms (Shao et al., 2024; Yu et al., 2025b), reasoning-oriented image generation and editing have achieved remarkable progress (Zhang et al., 2025b; Duan et al., 2025; Jiang et al., 2025; Wu et al., 2025b; Guo et al., 2025; Pan et al., 2025; Jin et al., 2024; Zhang et al., 2025a). Current methods may be classified according to the manner in which reasoning is incorporated into the pipeline: (i) *prompt interpretation*, where the system resolves compositional or implicit semantics in user instructions (*e.g.*, temporal or causal cues) prior to editing (Chen et al., 2025b; Sun et al., 2025; Jin et al., 2024; Zhang et al., 2025a); (ii) *prompt extension*, which augments concise instructions with additional structure (*e.g.*, constraints, spatial hints) to enhance output faithfulness (Wu et al., 2025b; Jiang et al., 2025; Zhang et al., 2025b; Duan et al., 2025); and (iii) *generation-time reasoning*, which introduces self-checking or iterative refinement during synthesis to enforce consistency with requirements (Guo et al., 2025; Pan et al., 2025). Nevertheless, these approaches are predominantly built on the assumption of a single, flattened canvas and thus lack *layer-aware* reasoning about hierarchical structure, inter-layer dependencies, and document-level constraints (*e.g.*, text fidelity, non-occluding layout). As a result, even when instructions are correctly interpreted, edits often fail to account for relevant layers or disrupt spatial organization. We introduce **MiLDEAgent**, which formalizes *multi-layer document editing* as a reasoning task and ensures consistency via layer selection, layer-wise editing instruction generation, and layer editing.

## 3 MiLDEBENCH

### 3.1 PRELIMINARIES

We define multi-layer document editing as a two-stage process consisting of reasoning and editing. A document $D$ is represented as an ordered set of transparent layers $\mathcal{L} = \{L_i \in \mathbb{R}^{H \times W \times C}\}_{i=1}^n$, rendered by alpha compositing $D = L_1 \oplus \cdots \oplus L_n$. Given a document-level instruction $I_D$, the reasoning stage is performed by a VLM-based reasoner $\mathcal{R}_\phi(D, I_D) \mapsto \hat{\mathcal{I}} = \{\hat{I}_i\}_{i=1}^n$, which predicts layer-specific instructions where each $\hat{I}_i$ either specifies an edit for layer $L_i$ or is a `no-op` indicating that the layer should remain unchanged. The editing stage is handled by an image-generation editor $\mathcal{E}(\mathcal{L}, I_D, \hat{\mathcal{I}}) \mapsto D'$, which updates the document by applying $L'_i = \mathcal{E}(L_i, \hat{I}_i)$ if $\hat{I}_i \neq$ `no-op`, and $L'_i = L_i$ otherwise. The final edited document is then reconstructed in the original order as $D' = L'_1 \oplus \cdots \oplus L'_n$. A valid solution must satisfy *instruction compliance* (the output follows the semantics, text, and attributes of $I_D$), *structural fidelity* (the global layout and all non-target content remain intact), and *layer awareness* (all and only the layers in $S^\star$ are modified). For diagnostic evaluation, the benchmark provides gold supervision in the form of $S^\star$ and $\mathcal{I}$, enabling measurement

of both document-level success (instruction following and fidelity) and decision quality (correctness of layer selection and alignment). Each benchmark instance is therefore specified by five components: the rendered document $D$, its layer decomposition $\mathcal{L}$, the document-level instruction $I_D$, the gold relevant-layer set $S^\star$, and the layer-wise instructions $\mathcal{I}$.

Since current open- and closed-source[1] models do not support multi-image (multi-layer) editing interfaces, we design a practical evaluation protocol that treats each method as a *black-box* editor.
Specifically, the model only consumes the rendered document $D$ and instruction $I_D$, and produces an edited output $D'$; layer-wise inputs or edits are *not* required. Even under this simplified setting, existing models fail to reliably follow instructions, preserve layout, or render texts (Table 2), underscoring the importance and difficulty of the proposed task: no previous work can fully complete it. Finally, Table 1 summarizes the dataset statistics. We also show the distribution of layers per document and prompt lengths in Figure 5 and Figure 6 in the Appendix.

Table 1: Statistics of MiLDEBench.

| Aspect | Train | Test |
|---|---|---|
| Number of design documents | 17.7k | 1.9k |
| Avg. #layers per doc | 4.45 | 4.44 |
| Avg. #layers needing edit per doc | 1.66 | 1.66 |
| Avg. len of doc-level instruction | 15.56 | 15.53 |
| Avg. len of layer-wise instruction | 24.50 | 24.48 |

### 3.2 DATASET CONSTRUCTION PIPELINE

Alg. 1 illustrates the overall data creation pipeline, including both document-level and layer-wise instruction generation.

**Design document collection and layer consolidation.** We build our corpus from the public Crello dataset (Yamaguchi, 2021), which provides transparent-background, multi-layer *design* documents represented as $(D, \mathcal{L})$, where $D$ is the rendered document and $\mathcal{L} = \{L_i\}_{i=1}^n$ is its layer decomposition. Crello is chosen because (i) our benchmark targets real-world design workflows with non-expert users, so we exclude datasets with synthetically generated layers (*e.g.*, Magick (Burgert et al., 2024), Prism-Layers (Chen et al., 2025a)); and (ii) our focus is on scenarios where text, decorative elements, and imagery interact, so we omit multi-layer resources derived from *natural* images (*e.g.*, MuLAn (Tudosiu et al., 2024), MLCID (Huang et al., 2024)). Although ART (Pu et al., 2025) introduces a large-scale design corpus, it is not publicly available and thus excluded. To make $\mathcal{L}$ tractable, we apply a *structure-preserving consolidation* procedure $\mathcal{C}(\mathcal{L}) \mapsto \mathcal{L}'$: an MLLM (InternVL3-38B (Zhu et al., 2025)) classifies layers into *text*, *decoration*, or *image*, and non-overlapping layers within each category are merged using layout metadata while preserving $z$-order and alpha boundaries. This reduces $|\mathcal{L}|$ (originally 2–50) to a semantically coherent $\mathcal{L}'$ without discarding content.

---

**Algorithm 1:** Data Construction Pipeline

**Input** : Design document $D$ with layers $\mathcal{L}$
**Output** : Validated document-level instruction $I_D$, layer-wise instructions $\mathcal{I} = \{I_i\}$, edited layers $S^\star$

**Part A: Document-level Instruction Generation**
1. Generate candidate instructions $\{I_D^j\}$ from $D$ via personas $p_j \sim$ PersonaHub;
2. Rank and filter $\{I_D^j\}$ by clarity, realism, and consistency;
3. Human validation $\Rightarrow$ finalize $I_D$.

**Part B: Layer-wise Instruction Generation**
1. Decompose $I_D$ into step-wise edits $\mathcal{A} = \{a_j\}$;
2. Match each $a_j$ to candidate layers $L_k \in \mathcal{L}$ using content-aware alignment;
3. Form preliminary instructions $I_k$ and filter by clarity, feasibility, and consistency;
4. Human validation $\Rightarrow$ finalize $\mathcal{I}$ and relevant-layer set $S^\star$.

---

**Document-level instruction generation.** Given a consolidated design document $(D, \mathcal{L})$, we generate a document-level instruction $I_D$ capturing both *content* and *layout* edits. For *content*, we adopt a two-stream pipeline that balances diversity and realism. (i) *Persona-based stream:* six personas $p_j \sim$ PersonaHub are sampled, and InternVL3-38B generates candidate instructions $I_D^{(j)}$ by adapting $D$ to each persona's domain while preserving its design intent (*e.g.*, "concert poster" $\rightarrow$ "historical exhibition poster"). (ii) *Document-based stream:* the model proposes semantically proximal domain transfers grounded in $D$ itself (*e.g.*, "summer camp" $\rightarrow$ "winter camp"). The combined candidate pool $I_D^{(j)}$ is then ranked by clarity, specificity, and realism, with low-quality cases removed through lightweight automatic filtering and regeneration until criteria are met. Finally, a human-in-the-loop validation stage ensures applicability and removes instructions that are infeasible or brand-violating, yielding the final $I_D$. For *layout*, InternVL3-38B generates six candidates spanning *move*, *swap*,

---

[1]We verified that GPT-o3 could complete the task in manual trials, but the model was discontinued before our benchmark was finalized, preventing systematic evaluation.

*scale*, and *rotate*; the two most substantive are selected through the same automatic filtering followed by human-in-the-loop review.

**Layer-wise instruction generation.** For each benchmark instance $(D, I_D, \mathcal{L})$, we provide a set of *layer-aligned* editing instructions $\mathcal{I} = I_i$ specifying how each relevant layer should be modified to realize the document-level intent. For *content editing*, during document-level instruction synthesis, the InternVL3-38B is simultaneously prompted to produce step-wise edits as a program that decomposes $I_D$ into atomic actions (*e.g.*, "replace text "piano concert" with "historical exhibition"", "swap the main image to a museum scene"). We then align steps to layers using a novel MLLM-based content-aware matcher to produce layer-wise instructions $I_i$. The matching algorithm is detailed in App. A.1.1. For *layout editing*, we introduce the details in App. A.1.2. Finally, automatically generated instructions are filtered by rule-based validators and refined through human-in-the-loop expert review, ensuring clarity, feasibility, and faithfulness to real design workflows. The resulting edited layers $S^\star$ and aligned instructions $\mathcal{I}$ thus combine automated alignment with human refinement to provide reliable gold supervision.

# 4 BENCHMARKING WITH MiLDEBENCH

## 4.1 MiLDEEVAL

For comprehensive evaluation on our benchmark, we propose **MiLDEEval**. For *content editing*, we propose four evaluation criteria: instruction following, layout consistency, aesthetics, text rendering and layer decision accuracy. For *layout editing*, we propose two evaluation criteria: instruction following and content consistency. We will introduce content editing evaluation in this section and layout editing evaluation in Appendix B.2.

**Instruction Following.** To assess whether the model faithfully executes an editing instruction $I_D$, we design a VQA-style evaluation metric. Given the document $D$, the target layer $S^\star$, and its layer-specific prompt $\mathcal{I}$, InternVL3-38B is prompted to generate a question–answer pair for each edited layer. Each question explicitly grounds the edit in spatial, textual, or entity-level detail (*e.g.*, *"Has the main image be changed to a museum scene?"*), with a binary answer of "yes" or "no." The instruction-following score is defined as the proportion of edits judged correct across all layers.

**Layout Consistency.** To evaluate structural fidelity, we measure layout consistency between original and edited documents using mask-level representations. We extract spatial masks $\mathcal{M} = \{M_i\}$ and $\mathcal{M}' = \{M_j'\}$ using Adopd Doc2Mask model (Gu et al., 2024) from the original document $D$ and edited document $D'$, then we design a new matching algorithm to match the two sets of spatial masks. For matched pairs, we assess position consistency (normalized centroid displacement), shape consistency (IoU), and area consistency (size ratio). Unmatched layers incur area-proportional penalties, with deleted layers penalized more heavily than newly created ones. The final score combines matching rate, average consistency scores, and penalty deductions with empirically tuned weights, providing a comprehensive measure of layout preservation robust to structural variations. The detailed calculation function is shown in App. B.1.1.

**Aesthetics.** We assess whether edits preserve or improve overall visual appeal using a frozen aesthetics predictor (*Aesthetic Predictor V2.5* (aes)). We directly utilize the score as final evaluation.

**Text Rendering.** We evaluate the *faithfulness* of edited text with an OCR–VQA pipeline. Specifically, we first apply the Adopd Doc2BBox model (Gu et al., 2024) to detect text regions in the edited image $L_j'$, and then use an MLLM to extract the corresponding text $t'$. Given the instruction $I_D$, we prompt the MLLM to verify whether $t'$ satisfies the required edit, producing a score in $\{0, 0.5, 1\}$. Unlike conventional text-alignment metrics (*e.g.*, SentenceBERT (Reimers & Gurevych, 2019)), our approach does not assume a unique ground truth: multiple valid edits may satisfy $I_D$, and thus a judgment-based evaluation better captures instruction faithfulness.

## 4.2 EVALUATION AND ANALYSIS

To conduct evaluation on **MiLDEBench**, we conduct comprehensive evaluation on 14 open-source models, with 12 reasoning-free models and 2 reasoning-enhanced models, and 2 closed-source models. Note that in these experiments, we only take design document $D$ and document-level editing

Table 2: Evaluation results of different models on content and layout editing tasks. For all scores, higher values indicates better performance. The highest score for closed-source and open-source text-to-image models are marked in red and blue respectively, and underline represents the second in open-source models.

| # | Model | Content Editing | | | | Layout Editing | |
|---|-------|------------------------|---------------------|------------|-------------------|------------------------|---------------------|
| | | Instruction Following | Layout Consistency | Aesthetics | Text Rendering | Instruction Following | Content Consistency |
| *Open-source Models* | | | | | | | |
| 1 | Instruct-Pix2Pix (Brooks et al., 2023) | 2.30 | 93.46 | 4.23 | 17.16 | 41.07 | 59.04 |
| 2 | MagicBrush (Zhang et al., 2023a) | 7.37 | 72.08 | 3.68 | 16.60 | 50.50 | 31.37 |
| 3 | UniWorld-v1 (Lin et al., 2025) | 5.75 | 61.59 | 3.91 | 22.04 | 48.59 | 56.61 |
| 4 | ICEdit (Zhang et al., 2025c) | 2.28 | 64.60 | 3.42 | 18.25 | 18.21 | 10.13 |
| 5 | UltraEdit (Zhao et al., 2024) | 12.41 | 85.31 | 3.54 | 11.39 | 44.37 | 61.64 |
| 6 | AnyEdit (Yu et al., 2025a) | 6.51 | 56.73 | 3.96 | 21.83 | 49.72 | 56.24 |
| 7 | OmniGen (Xiao et al., 2025a) | 3.83 | 85.96 | 3.90 | 19.76 | 51.10 | 57.30 |
| 8 | Qwen-Image-Edit (Wu et al., 2025a) | 10.09 | 74.20 | 4.12 | 24.32 | 59.25 | 71.35 |
| 9 | Flux1-Kontext (Batifol et al., 2025) | 12.49 | 48.32 | 3.94 | 19.31 | 60.84 | 64.31 |
| 10 | VAREdit (Mao et al., 2025) | 6.60 | 68.10 | 3.18 | 9.49 | 65.49 | 58.19 |
| 11 | Step1X-Edit (Liu et al., 2025) | 6.56 | 84.09 | 3.98 | 18.70 | 52.06 | 62.11 |
| 12 | Bagel (Deng et al., 2025) | 14.23 | 48.59 | 3.54 | 13.49 | 78.04 | 67.19 |
| *Reasoning-enhanced Models* | | | | | | | |
| 13 | Step1X-Edit w/ Thinking | 10.48 | 82.16 | 4.11 | 28.67 | 64.33 | 67.43 |
| 14 | Bagel w/ Thinking | 13.60 | 60.91 | 3.65 | 14.51 | 66.64 | 69.93 |
| *Closed-source Models* | | | | | | | |
| 15 | GPT-Image-1 (OpenAI, 2025) | 25.46 | 36.24 | 4.66 | 39.67 | 81.52 | 83.70 |
| 16 | Nano Banana (DeepMind, 2025) | 24.04 | 58.42 | 4.52 | 40.32 | 71.10 | 79.72 |

instruction $I_D$ as input, because current models cannot conduct multiple layer editing sinmutaneously. Specifically, the task here is $\mathcal{E}(D, I_D) \to D'$. The primary results are presented in Table 2. Please refer to Appendix C for detailed model setup.

**Finding 1: Current image editing models struggle with design document editing.** Our evaluation reveals that both open-source and closed-source models exhibit certain limitations in instruction following and text rendering. For open-source models (#1-#14), the average instruction-following accuracy is only about 10%, meaning that in nearly 90% of cases the specified edits are not correctly executed. Even the strongest closed-source baseline, GPT-Image-1 (#15), achieves only 25.46% instruction following accuracy, underscoring the substantial gap between current image editing capabilities and the demands of multi-layer document editing in realistic scenarios.

**Finding 2: Closed-source models achieve stronger instruction following but sacrifice format consistency.** Closed-source models substantially outperform open-source ones in instruction following, text rendering, and aesthetics. For example, in terms of instruction-following accuracy for content editing, GPT-Image-1 (#15) surpasses the best-performing open-source model Bagel (#12) by 78% (25.46% vs. 14.23%). For text-rendering score in content editing, Nano Banana (#16) exceeds the best-performing open-source model Step1X-Edit w/ Thinking (#13) by 40.6% (40.32% vs. 28.67%). However, these gains come at the expense of layout-consistency. In particular, GPT-Image-1 (#15) achieves the lowest score in layout-consistency, and Nano Banana (#16) performs only on par with the open-source average. For more examples, please refer to Appendix C. Notably, the comparably high layout-consistency scores in open-source models often stem from trivial artifacts, such as outputting the unedited document, which preserves layout without satisfying the instruction. This highlights a critical trade-off: closed-source models follow instructions more reliably, but they lack the ability to maintain structural fidelity in design documents, which is a limitation with significant consequences for real-world editing workflows.

**Finding 3: Reasoning-enhanced models provide only marginal gains for document editing.** Augmenting open-source editors with explicit reasoning mechanisms ("w/ Thinking") yields limited improvements. Step1X-Edit w/ Thinking (#13 vs. #11) improves instruction-following accuracy from 6.56% to 10.48% and achieves the highest text-rendering score (28.67%), suggesting that reasoning can help decompose instructions into more precise edits. However, Bagel w/ Thinking

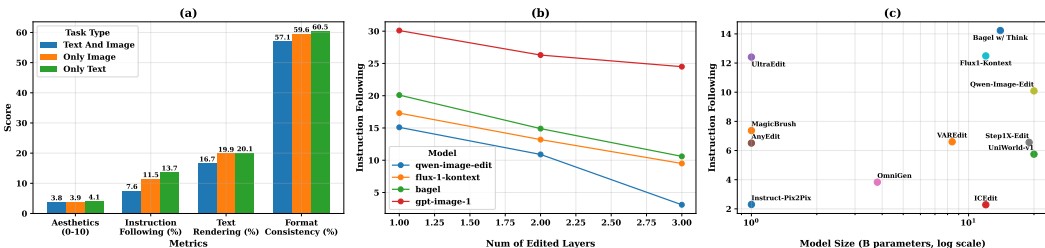

Figure 2: (a) Evaluation metrics with editing type. (b) Instruction following score with number of edited layers. (c) Instruction following score with model size.

(`#14` vs. `#12`) decreases instruction-following accuracy from 14.23% to 13.60% and provides no substantial gains in other metrics. Overall, the benefits remain modest relative to the difficulty of the task. Current reasoning modules primarily capture textual intent but struggle to ground edits within multi-layer document structures, especially when document-level editing prompts usually represents editing text and image sinmutaneously. This underscores the need for deeper multimodal reasoning integration, rather than shallow textual planning, to advance design document editing.

**Finding 4: Complex reasoning paths exacerbate editing errors.** Model performance degrades markedly as editing complexity increases. First, we sampled 150 cases from test set and classify them into three types based on the editing domain: text-only, image-only, and text+image editing. We report the average content editing score of three open-source models (Qwen-Image-edit, Flux1-Kontext, and Bagel). As shown in Figure 2 (a), instruction-following drops from 13.7% (text-only) and 11.5% (image-only) to 7.6% (text+image), with parallel declines in text rendering, aesthetics, and format consistency. Figure 2 (b) further reveals a strong effect of layer depth: Bagel falls from 20.1% (one layer) to 10.6% (three layers), Flux1-Kontext from 17.3% to 9.5%, and Qwen-Image-Edit from 15.1% to 3.1%; even GPT-Image-1 drops from 30.1% to 24.5%. Finally, Figure 2 (c) shows that larger model size does not consistently improve performance. In summary, performance degrades as editing complexity increases—both across modalities and with deeper layer structures—highlighting that current models struggle to reason over complex editing intents. Moreover, scaling model size does not consistently yield improvements, suggesting that advancing multimodal *reasoning capability* is crucial for progress in design document editing.

## 5 THE MILDEAGENT FRAMEWORK

Recognizing the reasoning inaccuracies, layout consistency issue and the fundamental problem that current image editing model cannot do multiple layer editng, we propose **MiLDEAgent**, consisting of an RL-trained reasoner and a freezed editor. Specifically, our agent receives a design document $D$ with multiple transparent background layers $\mathcal{L}$ and a document-level instruction $I_D$, and then produce $D'$ by editing *exactly* the relevant layers and re-compositing them in the original $z$-order. Specifically, the task here is $Agent(D, I_D, \mathcal{L}) \rightarrow (D', \mathcal{L}')$. We introduce our agent in Section 5.1 and evaluate on our benchmark on Section 5.2.

### 5.1 REASONING-GUIDED MULTI-LAYER DOCUMENT EDITING

Our **MiLDEAgent** is a two-stage framework for multi-layer document editing, where the reasoner $\mathcal{R}_\phi$ performs instruction decomposition and the editor $\mathcal{E}$ performs layer-wise editing.

**Reasoning.** The reasoning stage is handled by a VLM-based reasoner $\mathcal{R}_\phi$, which takes $(D, L_i, I_D)$ as input and outputs for each layer a binary decision $y_i \in \{0, 1\}$ and, if $y_i = 1$, a *layer-conditioned prompt* $I_i$. To train $\mathcal{R}_\phi$, we adopt Group Relative Policy Optimization (GRPO) (Shao et al., 2024), a RL method that evaluates groups of sampled responses, computes relative advantages by normalizing their rewards, and applies a clipped KL-regularized objective. This design reduces variance in credit assignment and encourages the model to distinguish between relatively better and worse responses, which is particularly beneficial for structured reasoning tasks (see Appendix D.1 for details).

Following this paradigm, we design a task-specific per-layer reward to supervise $\mathcal{R}_\phi$. The outputs of the reasoner must follow a structured format:

$$\texttt{<think>...</think> <decision>...</decision> <prompt>...</prompt>} \quad (1)$$

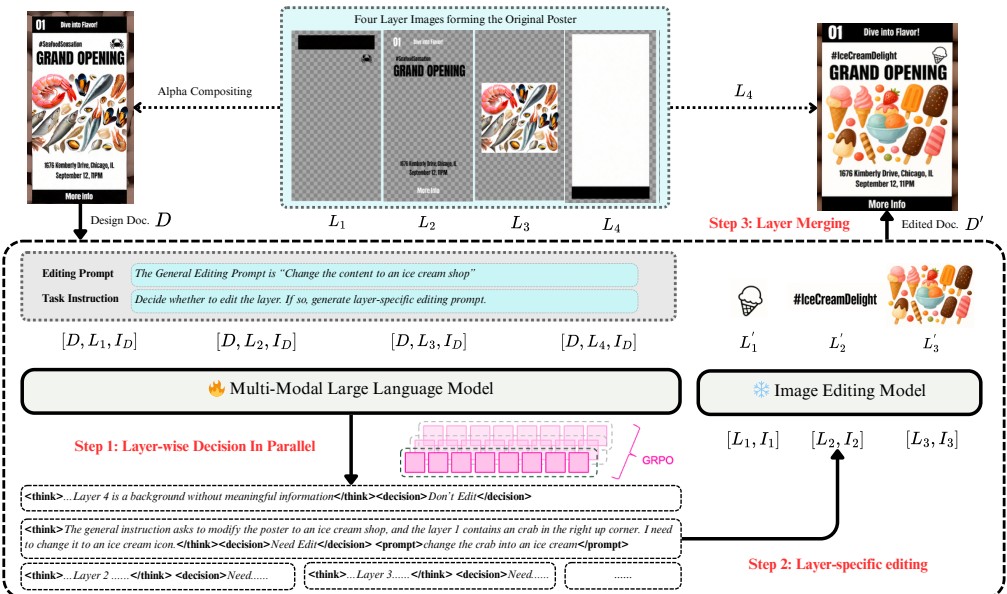

Figure 3: The illustration of MiLDEAgent.

where the three segments denote hidden reasoning, the binary decision $y_i$, and the layer-conditioned prompt $I_i$, respectively. The per-layer reward $\mathcal{R}_i$ then consists of three components:

$$r_f = \mathbb{1}[\text{format is valid}], \quad r_d = \mathbb{1}[y_i = y_i^\star], \quad r_p = \text{BLEU}(I_i, I_i^\star) \in [0, 1]. \tag{2}$$

The final per-layer reward is defined as

$$\mathcal{R}_i = \begin{cases} (r_f + r_d + r_p)/3, & r_d = 1, \\ (r_f + r_d)/2, & r_d = 0 . \end{cases} \tag{3}$$

where $r_f$ verifies syntactic correctness, $r_d$ measures decision accuracy against the gold label $y_i^\star = \mathbb{1}[L_i \in S^\star]$, and $r_p$ evaluates prompt quality relative to the reference instruction $I_i^\star$. The prompt reward $r_p$ is only applied when the decision is correct ($r_d = 1$).

**Editing.** The editing stage uses a frozen image-generation editor $\mathcal{E}$ for stability and modularity. For each selected layer $L_i$ ($y_i = 1$), a binary mask $M_i$ is extracted from its alpha channel (optionally refined with region cues), and the editor updates it as $L_i' = \mathcal{E}(L_i, I_i, M_i)$. For non-selected layers ($y_i = 0$), no operation is applied and $L_i' = L_i$. Transparency is preserved by restoring the original alpha to unedited regions. The final document is reconstructed by alpha compositing $D' = L_1' \oplus L_2' \oplus \cdots \oplus L_n'$, where $\oplus$ denotes standard alpha blending, ensuring global layout consistency while fulfilling the document-level instruction $I_D$.

## 5.2 EXPERIMENTAL RESULTS

**Setup.** We incorporate one of the SOTA MLLM, `QwenVL2.5-3B/7B` (Bai et al., 2025) as our reasoner, and applied GRPO algorithm to train on content editing tasks, with a freezed `Flux-1-Kontext` as editing model. The rollout number is set to 5 and the batch size to 512. All experiments are conducted on 8 A100 GPUs. In this section, we only train and evaluate on content editing setting which requires more reasoning capabilities due to the complex modality editing requirements. We will incorporate layout editing into our agent for future version. For evaluation, same as the four dimensions as introduced in Section 4.1, we add another dimension *Layer Decision Accuracy*, which represents whether our agent can correctly choose which layer should be edited.

**Quantitative Results.** As shown in Figure 4 (a), our proposed MiLDEAgent significantly outperforms all baselines in the content editing regime (Table 2). Specifically, MiLDEAgent achieves 20.7% in instruction following, representing a +50% improvement over the strongest open-source baseline (Bagel, 14.2%) and narrowing the gap with closed-source systems while preserving editability. On

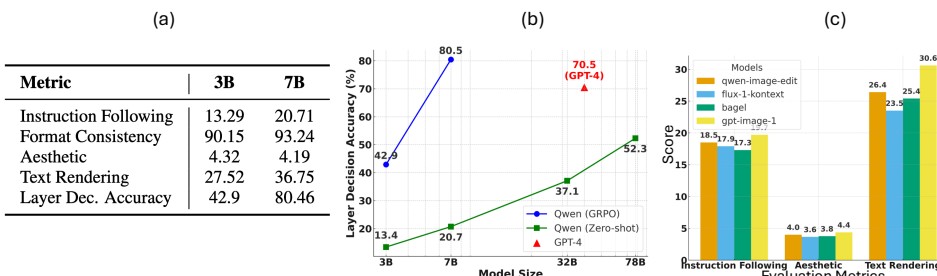

Figure 4: (a) The performance on content editing evaluation set. (b) Layer decision accuracy with model size. (c) The performance changes with different image editing models.

format consistency, MiLDEAgent maintains 93.2%, rivaling the best-performing diffusion-based editors and exceeding closed-source models by over +30 points. Importantly, our agent exhibits strong text rendering performance (36.8%), surpassing all open-source baselines ($\leq$ 24.3%) and approaching commercial systems (40%). This highlights the effectiveness of our reasoning-based approach in handling multi-layer textual elements, a persistent weakness of prior methods. Finally, on layer decision accuracy (80.5%), MiLDEAgent demonstrates robust layer-aware reasoning, an ability entirely absent from existing baselines, thereby validating the necessity of reasoning-enhanced frameworks for this task. Taken together, these results establish that multi-layer document editing requires explicit reasoning mechanisms, rather than relying solely on generation or editing heuristics. MiLDEAgent consistently balances instruction fidelity, fine-grained textual rendering, and layer-aware decomposition, making it the first system to robustly address multi-layer editing at scale.

**Ablation 1: GRPO-trained reasoner outperforms all zero-shot models in layer decision accuracy.** Reasoner is the key of MiLDEAgent, therefore, we conduct ablation study on the RL-trained reasoner with other larger open-/closed-source MLLMs on layer decision accuracy metrics. As shown in Figure 4 (b), we observe that models equipped with a GRPO-trained reasoner consistently surpass their zero-shot counterparts across all tested scales. For instance, QwenVL2.5-7B with GRPO achieves 80.5% accuracy, compared to only 20.7% for its zero-shot variant, a nearly 4× improvement. Similarly, QwenVL2.5-3B with GRPO improves from 13.4% to 42.9%, highlighting that structured reinforcement-style reasoning is beneficial even at smaller scales. Strikingly, the 7B GRPO-trained model not only outperforms all zero-shot baselines—including much larger 32B and 78B models—but also slightly outperforms GPT-4. These results underscore that *reasoning-oriented training, rather than model scaling alone, is the dominant factor for reliable layer decision making*, establishing GRPO as a crucial ingredient for advancing multi-layer document editing.

**Ablation 2: Image editing model also influence the final performance.** In this experiment, we randomly select 100 samples from content editing test set and utilize the GRPO-trained QwenVL2.5-7B model as reasoner to test different image editing models. As shown in Figure 4, although all models achieve broadly comparable scores, systematic differences emerge across evaluation dimensions. GPT-Image-1 consistently achieves the best overall results, with 19.7% in instruction following, 4.4% in aesthetics, and 30.6% in text rendering, outperforming the best open-source alternatives by a clear margin. Among open-source models, Qwen-Image-Edit exhibits relatively stronger instruction following and text rendering, while Bagel and Flux1-Kontext are more balanced but weaker in fidelity and reasoning. These results indicate that even with the same reasoning mechanism, the fidelity and controllability of the editing backbone strongly shape the final quality of document editing. Consequently, improvements in low-level editing architectures are complementary to reasoning-based approaches, and both are required to achieve robust performance in multi-layer editing.

# 6 CONCLUSION

In this work, we introduced MiLDEBench, the first benchmark for reasoning-based multi-layer poster editing, together with a novel evaluation metrics. Through comprehensive experiments, we demonstrated that existing methods struggle to accurately edit posters based on general simple editing prompt. To address these limitations, we proposed MiLDEAgent, which leverages a GRPO-trained reasoner for layer selection and prompt generation, coupled with a open-source image editor, significantly improving reasoning ability and editing quality.

ETHICS STATEMENT

This work introduces both a benchmark (MiLDEBench) and a reasoning-based editing agent (MiLDEAgent). The MiLDEBench is derived from the publicly available Crello dataset (Yamaguchi, 2021), and processed in compliance with their licenses. We did not create new visual content, instead, we extended the original Crello documents with general edit instructions and layer-wise edit instructions. No personally identifiable information or private user data is included in the benchmark. The MiLDEAgent is built entirely upon public models. We did not train any image editing model, but instead used existing open-source image editing models in a zero-shot manner. The only trained component is a reasoning model that outputs edit decisions and layer-specific instructions. Our contribution is intended solely to advance research on multi-layer document editing, and we do not anticipate direct misuse given the neutral and design-oriented nature of the data.

REPRODUCIBILITY STATEMENT

Our work consists of two main contribution, a novel benchmark and an agent-based solution. For data generation part, we utilize open-source public dataset and model, which is easy to be reproduced. For agent-based solution, we introduce the detailed reward design, algorithm choice and training parameters. We will also release our code in a short future.

THE USE OF LLMS

We use InternVL3-38B to generate and filter editing instructions as an extension of Crello documents, allowing precise, context-sensitive modifications across layers. To ensure clarity and fluency in textual content, we employ ChatGPT to polish writing.

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

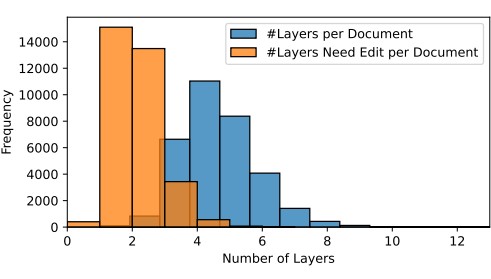

(a) Train set #layer distribution.

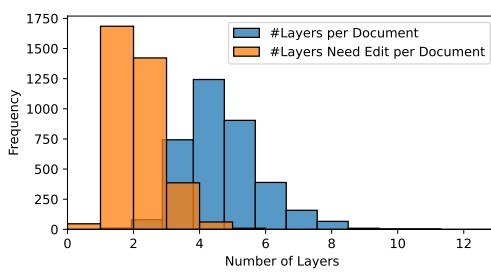

(b) Test set #layer distribution.

Figure 5: Distributions of the total number of layers per document and the number of layers requiring edits per document in the MiLDEBench.

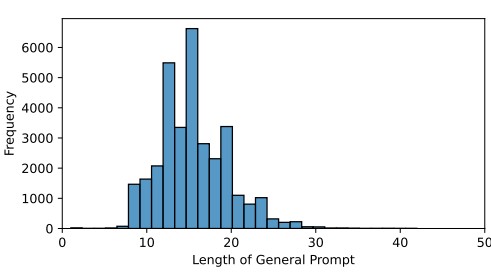

(a) Distribution of general prompt length.

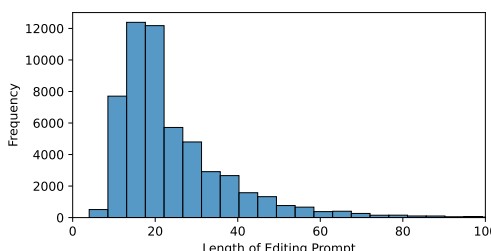

(b) Distribution of editing prompt length.

Figure 6: Distributions of general prompt lengths and the editing prompt lengths in the MiLDEBench.

Zechuan Zhang, Ji Xie, Yu Lu, Zongxin Yang, and Yi Yang. In-context edit: Enabling instructional image editing with in-context generation in large scale diffusion transformer. *arXiv preprint arXiv:2504.20690*, 2025c.

Haozhe Zhao, Xiaojian Shawn Ma, Liang Chen, Shuzheng Si, Rujie Wu, Kaikai An, Peiyu Yu, Minjia Zhang, Qing Li, and Baobao Chang. Ultraedit: Instruction-based fine-grained image editing at scale. *Advances in Neural Information Processing Systems*, 37:3058–3093, 2024.

Jinguo Zhu, Weiyun Wang, Zhe Chen, Zhaoyang Liu, Shenglong Ye, Lixin Gu, Hao Tian, Yuchen Duan, Weijie Su, Jie Shao, et al. Internvl3: Exploring advanced training and test-time recipes for open-source multimodal models. *arXiv preprint arXiv:2504.10479*, 2025.

# A DATA GENERATION PIPELINE

## A.1 LAYER-WISE INSTRUCTION GENERATION

### A.1.1 CONTENT EDITING

In this section, we describe the matcher used to align step-wise editing prompts with document layers. Given a set of step-wise prompts $\mathcal{I}_k$ and the layer set $S_j$ with known types (textual or visual), we first classify each prompt $\mathcal{I}_k$ using InternVL3-38B into either a *text-editing* or an *image-editing* category. A prompt is considered eligible only for layers of the corresponding type (i.e., text prompts for textual layers, image prompts for visual layers). Within each category, we process prompts sequentially: for each $\mathcal{I}_k$, we traverse the candidate layers in $z$-order and query InternVL3-38B to assess whether $\mathcal{I}_k$ semantically applies to $S_j$. Upon a positive match, $\mathcal{I}_k$ is assigned to $S_j$, and the procedure advances to the next prompt. This iterative matching continues until all prompts have been assigned or no valid layer remains.

### A.1.2 LAYOUT EDITING

For *Layout edits*, given $I_D$ and the set of relevant layers $\mathcal{L}$, we derive geometric targets by conditioning the MLLM on each layer crop, its current bounding box, and neighboring context. The model predicts proposed boxes or transforms (move/scale/rotate/swap). To preserve global structure, we run an iterative conflict-resolution loop: when proposed boxes overlap or violate margin constraints, a deterministic geometric solver computes candidate displacements, and the MLLM is re-queried (with updated context) to select admissible adjustments for the least salient/more movable elements. The process terminates upon feasibility or at a fixed iteration cap.

## B MiLDEEval

### B.1 CONTENT EDITING EVALUATION

#### B.1.1 LAYOUT CONSISTENCY

**Layout Consistency Evaluation** To assess the **structural fidelity** requirement—specifically whether the edited document $D'$ preserves the spatial arrangement and geometric relationships of elements— we introduce a comprehensive layout consistency metric that operates on mask-level representations of document layers. Given the inherent challenges of multi-layer editing where the number of layers may change ($|\mathcal{L}'| \neq |\mathcal{L}|$) and layer correspondences may be disrupted due to editing operations, our evaluation framework employs a principled matching strategy followed by multi-dimensional consistency assessment.

**Mask Extraction and Matching.** For both the original document $D$ and edited document $D'$, we extract layer-wise masks $\mathcal{M} = \{M_i\}_{i=1}^{|\mathcal{L}|}$ and $\mathcal{M}' = \{M_j'\}_{j=1}^{|\mathcal{L}'|}$ respectively using Adopd Doc2Mask model Gu et al. (2024), where each mask $M_i \in [0,1]^{H \times W}$ represents the spatial footprint of layer $L_i$. To establish correspondences between original and edited layers, we formulate mask matching as a bipartite graph optimization problem: we compute a pairwise IoU similarity matrix $\mathbf{S} \in \mathbb{R}^{|\mathcal{L}| \times |\mathcal{L}'|}$ where $S_{ij} = \text{IoU}(M_i, M_j')$, then apply the Hungarian algorithm to find the optimal matching $\mathcal{P}^* = \arg\max_{\mathcal{P}} \sum_{(i,j) \in \mathcal{P}} S_{ij}$ subject to IoU threshold filtering ($S_{ij} \geq \tau_{\text{IoU}}$).

**Multi-Dimensional Consistency Assessment.** For each matched pair $(M_i, M_j') \in \mathcal{P}^*$, we evaluate three complementary aspects of layout preservation: (1) **Position consistency** measures centroid displacement normalized by image diagonal: $c_{\text{pos}}(M_i, M_j') = 1 - \frac{\|\text{centroid}(M_i) - \text{centroid}(M_j')\|_2}{\sqrt{H^2 + W^2}}$; (2) **Shape consistency** directly uses the IoU between masks: $c_{\text{shape}}(M_i, M_j') = \text{IoU}(M_i, M_j')$; (3) **Area consistency** computes the ratio of smaller to larger mask areas: $c_{\text{area}}(M_i, M_j') = \frac{\min(\text{area}(M_i), \text{area}(M_j'))}{\max(\text{area}(M_i), \text{area}(M_j'))}$.

**Unmatched Layer Penalty.** To account for layers that appear or disappear during editing, we introduce a penalty mechanism that distinguishes between disappeared layers (present in $\mathcal{L}$ but unmatched in $\mathcal{L}'$) and newly created layers (present in $\mathcal{L}'$ but unmatched in $\mathcal{L}$). The penalty for each unmatched layer is proportional to its normalized area, with disappeared layers receiving full penalty and new layers receiving a reduced penalty (coefficient 0.7) to reflect that layer creation may be intentional: $p_{\text{disappeared}} = \sum_{i \in \mathcal{U}_{\text{orig}}} \text{area}(M_i)$ and $p_{\text{new}} = 0.7 \sum_{j \in \mathcal{U}_{\text{edit}}} \text{area}(M_j')$, where $\mathcal{U}_{\text{orig}}$ and $\mathcal{U}_{\text{edit}}$ denote unmatched layer indices.

**Final Score Computation.** The overall layout consistency score aggregates matched-layer performance with unmatched-layer penalties:

$$\text{LayoutConsistency} = \max \Big( 0, \omega_{\text{match}} \cdot r_{\text{match}} + \omega_{\text{pos}} \cdot \bar{c}_{\text{pos}} + \omega_{\text{shape}} \cdot \bar{c}_{\text{shape}}$$

$$+ \omega_{\text{area}} \cdot \bar{c}_{\text{area}} - \omega_{\text{penalty}} \cdot \big( p_{\text{disappeared}} + p_{\text{new}} \big) \Big), \qquad (4)$$

where $r_{\text{match}} = \frac{|\mathcal{P}^*|}{\max(|\mathcal{L}|, |\mathcal{L}'|)}$ is the matching rate, $\bar{c}$. denotes average consistency scores across matched pairs, and $\{\omega\}$ are empirically set weights (0.25, 0.2, 0.2, 0.2, 0.15 respectively). This metric provides a comprehensive assessment of layout preservation that is robust to layer count variations and sensitive to both geometric distortions and structural changes.

## B.2 Layout Editing Evaluation

For layout editing evaluation, we also evaluate from two aspects: instruction following and content editing. Here, we also utilize a VQA-based metrics to ask an MLLM evaluator whether the edited document follows the editing instruction, or whether the edited document remains the content as the same. We use a binary "yes/no" as the final score. We acknowledge that such a VQA-based evaluation metrics is not quite reliable. We will propose a more comprehensive evaluation method in our future version.

## C Experiments

**Baseline Open-source Models** We evaluate on 14 open-source models covering auto regressive and diffusion-based framework. The model size ranges from 1B to 20B. In Table 3.

## D MiLDEAGENT

### D.1 Preliminary of GRPO Algorithm

Group Relative Policy Optimization (GRPO) (Shao et al., 2024) has been proved to be helpful for improving reasoning capabilities for LLM (Shao et al., 2024), Multi-modal understanding (Huang et al., 2025) and even image generation (Zhang et al., 2025b; Jiang et al., 2025). GRPO computes advantages from a group of responses. Given each question-anwer pair $(q, a)$, old policy $\pi_{\theta_{\text{old}}}$ randomly samples $G$ responses, denoted as $\{o_i\}_{i=1}^{G}$. Each response $o_i$ is then fed into a reward model to obtain a reward $R_i$. Then, the advantage of the $i$-th response is obtained by normalizing the rewards of the group:

$$A_i = \frac{\mathcal{R} - \text{mean}(\{\mathcal{R}_i\}_{i=1}^{G})}{\text{std}(\{\mathcal{R}_i\}_{i=1}^{G})} \tag{5}$$

GRPO applies a clipped objective similar to PPO with a KL penalty term:

$$\mathcal{J}_{\text{GRPO}}(\theta) = \mathbb{E}_{(q,a) \sim \mathcal{D}, \{o_i\}_{i=1}^{G} \sim \pi_{\theta_{\text{old}}}(\cdot|q)}$$

$$\left[ \frac{1}{\sum_{i=1}^{G} |o_i|} \sum_{i=1}^{G} \sum_{t=1}^{|o_i|} \left( \min \left( r_{i,t}(\theta)\hat{A}_i, \text{clip}\left(r_{i,t}(\theta), 1 - \varepsilon, 1 + \varepsilon\right)\hat{A}_i \right) - \beta D_{\text{KL}}\left(\pi_\theta || \pi_{\text{ref}}\right) \right) \right], \tag{6}$$

where $r_{i,t}(\theta)$ is the important weight for each token $t$:

$$r_{i,j}(\theta) = \frac{\pi_\theta\left(o_{i,j} \mid q, o_{i,<j}\right)}{\pi_{\theta_{\text{old}}}\left(o_{i,t} \mid q, o_{i,<j}\right)}. \tag{7}$$

Usually in the reasoning task with only textual output, the model is asked to generate responses following a structured format. The total rewards consists of two rule-based rewards: (1) format reward and the accuracy of the specific downstream task.

### D.2 MiLDEAGENT

**Failure cases of MiLDEAgent.** Our agent is not without failure modes. First, since layer decisions are made independently, multiple layers may occasionally be edited simultaneously, leading to unintended overlaps or conflicts that degrade final quality. Second, as shown in Figure 4(a), even when layer decision accuracy is high (e.g., with the 7B model), the overall instruction-following score remains low. This discrepancy can arise from two factors: (i) ambiguous or underspecified layer-wise editing prompts, and (ii) the inherent limitations of the underlying image editing model. A promising direction to mitigate these issues is to incorporate a self-checking mechanism that verifies the merged edited document and triggers regeneration when inconsistencies are detected. We leave such improvements for future work.

Table 3: Evaluation results for different models on content and layout editing tasks

| Model | Size | Type | Reasoning-Enhanced |
|---|---|---|---|
| Instruct-Pix2Pix Brooks et al. (2023) | 1B | Diffusion | ✗ |
| MagicBrush Zhang et al. (2023a) | 1B | Diffusion | ✗ |
| UniWorld-v1 Lin et al. (2025) | 20B | Diffusion | ✗ |
| ICEdit Zhang et al. (2025c) | 12B | Diffusion | ✗ |
| UltraEdit Zhao et al. (2024) | 1B | Diffusion | ✗ |
| AnyEdit Yu et al. (2025a) | 1B | Diffusion | ✗ |
| OmniGen Xiao et al. (2025a) | 3.8B | Diffusion | ✗ |
| Step1X-Edit Liu et al. (2025) | 19B | Diffusion | ✗ |
| Qwen-Image-Edit Wu et al. (2025a) | 20B | Diffusion | ✗ |
| Flux1-Kontext Batifol et al. (2025) | 12B | Diffusion | ✗ |
| Bagel w/o Think Deng et al. (2025) | 14B | Diffusion | ✗ |
| Bagel w/ Think Deng et al. (2025) | 14B | Diffusion | ✓ |
| VAREdit Mao et al. (2025) | 8.4B | AR | ✗ |
| DIM-Edit Zeng et al. (2025) | 4.6B | Diffusion | ✗ |

