# OpenReview forum: "MiLDEdit: Reasoning-Based Multi-Layer Design Document Editing"
_ICLR.cc/2026/Conference — ICLR 2026 Conference Withdrawn Submission_

### Official Review · Reviewer_dY1w · 2025-10-30

**Soundness:** 2
**Presentation:** 3
**Contribution:** 3
**Rating:** 4
**Confidence:** 4

**Summary:**

This paper focuses on the multi-layer design document editing task. A reasoning-based framework, i.e., MiLDEAgent, is proposed with an RL-trained multimodal reasoner and an image editor. The multi-layer document editing benchmark is introduced with 20k design documents paired with diverse editing instructions. Moreover, a task-specific evaluation protocol is used with 4 dimensions for content editing and 2 dimensions for layout editing. The experimental results show the effectiveness of the proposed method for multi-layer document editing.

**Strengths:**

This paper provides a comprehensive benchmark for the multi-layer design document editing task. Multi-dimensional metrics from content editing and layout editing aspects have been considered. The proposed benchmark and evaluation protocol could be useful for future research in the community. The proposed method is also a reasonable and effective solution for design document editing.

**Weaknesses:**

1. The statement of the proposed benchmark with two complementary axes (i.e., content editing and layout editing) is a little bit misleading. The main experimental analysis focuses almost exclusively on content editing, with layout editing results relegated to Appendix B.2. Moreover, based on the layout editing evaluation in Appendix B.2, the current evaluation metrics are not quite reliable (L768-L763).
2. The choice of Crello over synthetically generated datasets like PrismLayers (Chen et al., 2025a) is not fully convincing. Different from natural images, design documents are synthesized by designers following certain rules. As long as the data construction process adopts these rules, these high-quality design documents could also be used in the benchmark.
3. Technical novelty and performance gains of the proposed MiLDEAgent are overstated. The proposed system comprises a multimodal reasoner trained via GRPO and a pluggable layer-wise image editor, both of which build heavily on existing paradigms. In addition, based on the results in Table 2, the results do not surpass open-source baselines significantly as stated. For example, the layout consistency and aesthetics of the proposed method are lower than those Instruct-Pix2Pix (Brooks et al., 2023).

**Questions:**

1. Why layout editing of the proposed agent is not considered in the experiment (see L426)?
2. As both the document-level instruction generation and layer-wise instruction generation contain a human-in-the-loop validation stage, I was wondering about the details and the cost of the participants. How to ensure consistency among different participants?
3. How about the computational cost and running time analysis between the proposed method and existing open-source/closed-source models?

---

### Official Review · Reviewer_85sA · 2025-10-30

**Soundness:** 3
**Presentation:** 3
**Contribution:** 3
**Rating:** 4
**Confidence:** 3

**Summary:**

This paper introduces MiLDEBench, the first benchmark for reasoning-based multi-layer design document editing, along with MiLDEAgent, a novel framework that combines RL-trained multimodal reasoning with layer-wise editing. The benchmark comprises 20,000 transparent-background design documents from the Crello dataset, augmented with 50,000 natural-language editing instructions and 87,000 layer-aligned edit steps validated through human-in-the-loop processes. The authors propose MiLDEEval, a task-specific evaluation protocol measuring instruction following, layout consistency, aesthetics, text rendering, and content consistency. After benchmarking 14 open-source and 2 closed-source models, they demonstrate that existing approaches struggle significantly with multi-layer editing. Their proposed MiLDEAgent employs a GRPO-trained multimodal reasoner for layer selection and prompt decomposition, paired with a frozen image editor, achieving 50% improvement over open-source baselines and 80.5% layer decision accuracy.

**Strengths:**

1、 The paper addresses a genuine gap in current vision-language research by formalizing multi-layer design document editing, which has clear practical applications in real-world creative workflows.

2、MiLDEBench is carefully curated with human-in-the-loop validation, incorporating both document-level and layer-aligned instructions. The data generation pipeline using persona-conditioned and document-conditioned prompts ensures diversity and realism.

**Weaknesses:**

1、 MiLDEAgent is only trained and evaluated on content editing, not layout editing, which represents only half of the proposed benchmark. This significantly limits the contribution's completeness and practical applicability.

2、The reasoner makes per-layer decisions independently, which can lead to conflicting edits when multiple layers interact. This is a fundamental architectural limitation that could cause cascading errors in complex documents.

3、 While Figure 2 shows performance degradation with complexity, there's insufficient qualitative analysis of when and why the agent fails, which would be valuable for guiding future work.

**Questions:**

see weakness

---

### Official Review · Reviewer_iKdi · 2025-10-31

**Soundness:** 2
**Presentation:** 3
**Contribution:** 3
**Rating:** 6
**Confidence:** 3

**Summary:**

The paper studies the problem of multi-layer design document editing.

The paper contributes: 1) a benchmark (MiLDEBench) consisting of multi-layer design documents along with MLLM-generated document-level  and layer-level editing instructions; 2) a set of evaluation metrics (MiLDEEval) that cover different content and layout dimensions; 3) a reasoning-based editing method (MiLDEAgent) that leverages the reasoning abilities of a multimodal large language model to select layers to edit and synthesize layer-wise editing instructions, and then performs layer-wise editing with an image editor.

**Strengths:**

1. Multi-layer design document editing is a challenging and important problem to investigate.

2. The dataset construction pipeline is well designed, and the evaluation metrics are well defined. The resulting benchmark (MiLDEBench) and evaluation protocol (MiLDEEval) could encourage and benefit future works on design document editing.

3. The proposed reasoning-based editing method (MiLDEAgent) is well developed and shows promising results.

**Weaknesses:**

1. The experimental setup in Section 4.2 is questionable. All the existing models observe the input design document through only a rendered image. Thus, they are not aware of the design’s multi-layer structure, and are not given element attributes explicitly. A better option could be to provide the structure and attribute information directly to some of these models. For example, it is possible to create a structured representation (e.g., in JSON format) that contains the attributes (e.g., category, position, size, z-order, text) of each document element, and input it into the MLLM-based models. This would allow them to better understand the input design, thereby improving their editing results.

2. A user study comparing the proposed model and a subset of existing models included in Section 4.2 is missing, and should be added.

3. The visual comparison of edited results by different methods is not provided in the paper, and should be added.

**Questions:**

1. In Section 3.2, which layers of the design documents on Crello are classified as “decoration”?

2. In “Ablation 1” of Section 5.2, what is the reason for not reporting the results of different models on the four content editing metrics?

---

### Official Review · Reviewer_tA31 · 2025-11-03

**Soundness:** 3
**Presentation:** 1
**Contribution:** 2
**Rating:** 2
**Confidence:** 3

**Summary:**

The paper tackles the multi-layer design document editing task, where a new dataset and benchmark, MiLDEBench, are proposed to evaluate existing models and the GRPO fine-tuned Qwen model in this work.

**Strengths:**

1. The paper introduces a new benchmark targeting layered document editing, which is overlooked by prior works.

2. The proposed GRPO fine-tuning brings some improvements over the open-source baseline models on the proposed evaluation metrics.

**Weaknesses:**

1. Overclaimed performance improvement. The authors claim "Delivering over 50% improvements compared to all open-source baselines and attaining performance comparable to closed-source models" (L051).

Importantly, the proposed method is not directly compared to existing methods in Table 2, which is not a standard practice in a benchmark paper.

- If we take the 3B model result in Figure 4 (a) and compare it with open-source models in Table 2, Instruction Following (13.29 vs 14.23 BAGEL) / Layout Consistency (90.15 vs 93.46 Instruct-Pix2Pix) / Text Rendering (27.52 vs 28.67 Step1X-Edit w/ Thinking) are not better than the best model in the table.

- If we take the larger 7B model, the aesthetics score is even worse (4.19 vs 4.32 for 3B). Other metrics do improve, but instruction following (20.71), layout consistency (93.24, close to Instruct-Pix2Pix), and Text Rendering (36.75) are nowhere near over 50% improvements over all open-source baselines.

- L430 "Specifically, MiLDEAgent achieves 20.7% in instruction following, representing a +50% improvement over the strongest open-source baseline (Bagel, 14.2%)" - the improvement is 45.7%, not 50%


2. Incomplete method evaluation. L426 "We will incorporate layout editing into our agent for future version." As this paper is not proposing a new algorithm, but instead directly uses GRPO. For a benchmark and dataset paper, it is insufficient to leave out the layout editing part. It remains unclear if the proposed method works for that setting.


3. Poor presentation and unclear structure.
- It would be better to put the layout consistency part in the main paper instead of the appendix, while I find the findings in Section 4.2 pedantic.
- Table 3 caption is irrelevant. The table itself does not show any evaluation results.
- On page 9, there is "format consistency", while the metric was originally introduced as "layout consistency".
- Figure 5 and 6 are somewhere in the middle of the references.
- Algorithm 1 block seems unnecessary as the data curation process is already described in its surrounding paragraph.

**Questions:**

1. In Figure 4 (c), it appears Qwen-image-edit shows the best performance among the open-source models. Why did the authors not use it as the default editing model?

2.  L413 - a binary mask $M_i$ is extracted from its alpha channel (optionally refined with region cues). How does this optional refinement process work?

3. One major limitation I see is that if the edited layer introduces new alpha regions, the proposed framework will not be able to handle them, as the image editors used only operate on RGB channels. In L414 - "Transparency is preserved by restoring the original alpha to unedited regions". How about the edited regions? If we take the poster sample in Figure 1, the edited text "SOLVE THE MSEFSTERY" has a different shape than the original text, which means the original alpha mask will not apply. It is unclear how the authors derived the new alpha mask.

---

### Note · Authors · 2025-11-14

I have read and agree with the venue's withdrawal policy on behalf of myself and my co-authors.